# Clinical Impact of Vertical Artifacts Changing with Frequency in Lung Ultrasound

**DOI:** 10.3390/diagnostics11030401

**Published:** 2021-02-26

**Authors:** Natalia Buda, Agnieszka Skoczylas, Marcello Demi, Anna Wojteczek, Jolanta Cylwik, Gino Soldati

**Affiliations:** 1Department of Internal Medicine, Connective Tissue Diseases and Geriatric, Medical University of Gdansk, 80-952 Gdansk, Poland; wojteczkowa@vp.pl; 2Department of Geriatrics, National Institute of Geriatrics Rheumatology and Rehabilitation, 02-637 Warsaw, Poland; agnieszka.skoczylas@spartanska.pl; 3Department of Medical Image Procesing, Fondazione Gabriele Monasterio, 56124 Pisa, Italy; demi@ftgm.it; 4Department of Anaesthesiology and Intensive Therapy, Mazovia Regional Hospital in Siedlce, 08-110 Siedlce, Poland; jolacylwik@o2.pl; 5Interventional and Diagnostic Ultrasound Unit, Valle del Serchio, General Hospital Lucca, 55100 Lucca, Italy; gino.soldati@uslnordovest.toscana.it

**Keywords:** B-line, lung ultrasonography, LUS, interstitial lung disease, systemic sclerosis, pulmonary fibrosis, oedema

## Abstract

Background: This study concerns the application of lung ultrasound (LUS) for the evaluation of the significance of vertical artifact changes with frequency and pleural line abnormalities in differentiating pulmonary edema from pulmonary fibrosis. Study Design and Methods: The study was designed as a diagnostic test. Having qualified patients for the study, an ultrasound examination was performed, consistent with a predetermined protocol, and employing convex and linear transducers. We investigated the possibility of B-line artifact conversion depending on the set frequency (2 MHz and 6 MHz), and examined pleural line abnormalities. Results: The study group comprised 32 patients with interstitial lung disease (ILD) (and fibrosis) and 30 patients with pulmonary edema. In total, 1941 cineloops were obtained from both groups and analyzed. The employment of both types of transducers (linear and convex) was most effective (specificity 91%, specificity 97%, positive predictive value (PPV) 97%, negative predictive value (NPV) 91%, LR(+) 27,19, LR(−) 0.097, area under curve (AUC) = 0.936, *p* = 7 × 10^−6^). Interpretation: The best accuracy in differentiating the etiology of B-line artifacts was obtained with the use of both types of transducers (linear and convex), complemented with the observation of the conversion of B-line artifacts to Z-line.

## 1. Introduction:

Lesions affecting the interstitium are most frequently caused by pulmonary edema (cardiogenic and non-cardiogenic) and interstitial lung disease (ILD) [1]. Such lesions are easily detected in a lung ultrasound (LUS), where B-line artifacts are searched for. However, without considering clinical data, differentiating the etiology of lesions affecting the interstitium is much more difficult [2,3,4,5]. Consequently, searching for further possibilities for differentiating a pulmonary from cardiogenic etiology of interstitial lesions using LUS is well grounded.

In this study, vertical artifacts were analyzed (depending on the set operating frequency), as well as pleural line abnormalities. The first goal was to compare the length of vertical artifacts evaluated with a convex transducer at two extreme frequencies: 2 MHz, and then 6 MHz. The second goal was to assess pleural line abnormalities, with the employment of a linear transducer in both patient groups.

B-line artifacts are significant in diagnosing many diseases that affect the pulmonary interstitial space and alveoli [1,6]. These artifacts are defined as laser-like vertical reverberation artifacts arising from the pleural line, extending to the bottom of the screen (irrespective of the set depth), moving along with the lung slide, and leading to the disappearance of A-lines [7]. Z-line artifacts belong to one family of vertical artifacts, similar to B-line artifacts however, they are much shorter and do not extend to the bottom of the screen [8,9,10]. The mechanism of B-line and Z-line formation is still not fully examined.

## 2. Materials and Methods

### 2.1. Study Design

The study was conducted as a prospective cohort study. Approval from the local ethics committee (number: NKBBN/474/2018 and NKBBN/473/2018) and the informed consent of all participants in the study was duly obtained. Approval date for both 10 October 2018.

### 2.2. Study Population

Two groups of patients were examined: those patients diagnosed with ILD secondary to systemic sclerosis (group A), and patients diagnosed with pulmonary edema due to the exacerbation of congestive heart failure or to acute heart failure (group B). The exclusion criteria for patients with recognized ILD were as follows: comorbidity of congestive heart failure, pneumonia, and noncardiogenic edema. For patients diagnosed with pulmonary edema, the exclusion criteria were: pulmonary fibrosis, ILD, pneumonia, and noncardiogenic edema. The findings were anonymized and entered into a database by independent members of the research project. Written informed consent was obtained from those patients who agreed to participate. Duration of symptoms, examination findings, comorbidities, treatment, laboratory test results and echocardiography examination results, chest X-rays, and (in the case of ILD) high resolution computed tomography (HRCT) results were recorded. Patients were evaluated with LUS, and findings were recorded on standardized forms. 

### 2.3. Study Protocol

LUS examinations were performed by three independent operators who are clinicians experienced in sonography (4 years, 10 years, and 10 years). Ultrasound examinations were recorded and re-analyzed by clinicians and physics specialists. An ultrasonography device (Philips Sparq, made in Bothell, WA, USA, 2013), with a 2–6 MHz convex curved transducer, and a 4–12 MHz linear transducer, was used. Patients were evaluated with the application of LUS performed in the same manner, and with the same technical criteria: (a) speckle reduction, compound imaging, and tissue harmonic imaging were switched off; (b) the focus of the image was positioned at the pleural line level; (c) imaging depth was set at 15 cm for a convex transducer, and at 6 cm for a linear transducer; (d) gain and time gain compensation (TGC) were adjusted in mid-scale. Moreover, when the lungs were examined with a convex transducer, vertical artifacts were evaluated with two extreme frequencies: 2 MHz, and then 6 MHz. When a convex transducer was employed, the sonomorphology of all artifacts, in both groups, was compared to each other, and analyzed statistically. When a linear transducer was employed, pleural line abnormalities were evaluated. Sonographic examinations were performed in the supine position and through the intercostal spaces on both sides of the chest. The probes were applied at four points over the front of the chest, and eight points over the posterior–lateral part of the chest. 

### 2.4. Statistical Analysis

Data analyses were performed in R statistical software (open source (GNU license) statistical environment available with libraries (stat, pROC, plyr, ggplot2) at www.r-project.org (accessed on 26 February 2021)), version 3.6.0, using the following software: stat, plyr, ggplot2, pROC. The results were presented as the mean (standard deviation) for continuous variables and count (frequency) for discrete data. A *p*-value < 0.05 was regarded as statistically significant. Discrete data were compared for the groups with Pearson’s χ^2^ test, with appropriate modifications (i.e., Yates’s correction, Fisher’s exact test or V^2^ test). For ultrasonographic features differentiating pulmonary fibrosis from heart failure, independently and in complex models, a receiver operating characteristic (ROC) curve was plotted, and the area under the curve (AUC) was calculated, determining whether it differed statistically by 0.5 with the application of the DeLong test. AUCs for differentiating parameters and predictive models were compared with the DeLong test. For quasi continuous variables (e.g., a total number of intercoastal spaces containing consolidations) optimal cut-off points were determined with two methods (“closest topleft” and Youden). For all diagnostic parameters, sensitivity, specificity, positive predictive value (PPV), negative predictive value (NPV), and the likelihood ratio for a positive LR(+) and negative result LR(−) were calculated. For statistically significant models, logistic regression was performed, calculating the odds ratio for a positive result of the tested model and a respective Akaike information criterion (AIC) value. AIC allows for the comparison of different models, where the lower its value, the better a given model is adjusted to the experimental data. 

### 2.5. A Differentiating Model’s Assumption

A complete predictive model should take into account as many differentiating elements as possible in different chest areas and the examination technique. We considered three complex models: 

A—change in the image of the length of the vertical lines artifacts in three or more areas, and lack of consolidation as a fibrotic feature; 

B—change in the image of length of the vertical artifacts in three or more areas, lack of: consolidation, irregularity, fragmentation, or blurring of the pleural line in at least two points as a feature of pulmonary fibrosis.

C—change in the image length of the vertical artifacts in three or more areas, or irregularity, fragmentation, or blurring of the pleural line in at least two points, as a feature of pulmonary fibrosis.

All models were created by taking into account the ultrasound protocol: use of the convex transducer to visualize vertical artifacts, followed by a linear one and the assessment of the pleural line. Models were limited to the anterior and posterolateral areas of the chest, which makes it possible to use them in the diagnosis of severe conditions, where ultrasound evaluation of the posterior surface of the chest is impossible. All models require the assessment of vertical artifacts and the pleural line in at least six points: upper, lower, and posterolateral bilaterally. The statistical properties of these models are presented in Table 1. 

## 3. Results

### 3.1. Group A Characteristics—Patients with ILD

A total of 32 consecutive patients diagnosed with ILD secondary to systemic sclerosis qualified for the study, 17 females and 15 males, with an average age of 56 (21.2 SD) years. Diagnosis of systemic sclerosis was based on current ACR/EULAR 2013 criteria. ILD in this patient group was diagnosed on the basis of: patient’s clinical picture, abnormalities in immune tests, HRCT as well as pulmonary function tests, bronchofiberoscopy, and echocariography. Infection as the cause of lesions affecting the interstitium was excluded based on microbiological tests.

### 3.2. Group B Characteristics—Patients with Pulmonary Edema

30 consecutive patients with a clinical diagnosis of exacerbation of left ventricular failure and acute pulmonary edema qualified for the study, 13 females and 17 males, with an average age of 69 (21 SD) years. Pulmonary edema was diagnosed on the basis of clinical symptoms (dyspnea, orthopnea, bilateral abnormalities on auscultation: crackles), high NT-proBNP (N-terminal pro—brain natriuretic protein) level, and typical abnormalities indicating edema visible in a chest X-ray and echocardiography. 

### 3.3. Analysis of LUS Findings

During the study, 789 video clips (cineloops) were obtained and analyzed from patients with pulmonary edema, as well as 1152 cineloops from patients with ILD secondary to systemic sclerosis. While, 876 cineloops containing vertical artifacts evaluated with a convex transducer, and 644 cineloops assessing a specific area with a linear transducer were selected from the collected material, respectively. The recorded cineloops were assessed by clinicians and an engineer specializing in physics. Following the analysis of the video recordings, 128 cineloops were rejected due to inappropriate ultrasound device settings. The remaining cineloops were analyzed as regards the length of the artifacts (convex transducer), and determining whether a given artifact meets the definitional criteria of B-line (long artifact) or Z-line (short artifact). Moreover, the sonomorphology of the pleural line was analyzed in all patient, and evaluated with a linear transducer. 

### 3.4. LUS Findings—Convex Transducer:

B-line artifacts are detected in patients with cardiogenic pulmonary edema and ILD. In this study, B-line artifacts were evaluated with a convex transducer, at a depth of 15 cm, consistently with the settings described in the methodology section. The ultrasound frequency was changed during the examination, and the evaluation was performed at two extreme values: 2 MHz and 6 MHz.

In the cineloops obtained from patients with both pulmonary edema and ILD, B-line artifacts were almost always visualized at the frequency of 2 MHz (consistent with the adopted definition of B-line artifact), in 68% of the examined points in patients with pulmonary edema, and 63% of the examined points in patients with ILD, respectively. Z-lines were visible only in single points: one (0.4%) in heart failure, and 11 (3%) in pulmonary fibrosis, whereas in three cases (0.8%) Z-lines coexisted with B-lines. 

At the frequency of 6 MHz, cineloops recorded for patients with ILD presented Z-lines in 62% of the evaluated points and B-lines in 13%, whereas in 10% of the examined points the findings were mixed (Figure 1). 

In patients with cardiogenic pulmonary edema, at the frequency of 6 MHz, B-line artifacts were present in 62% of the evaluated points, and Z-lines in 24%, including the mixed profile of B and Z in 16% of the examined areas (Figure 2). Consequently, the change in frequency leads to a change in the profile of vertical artifacts, whereas this phenomenon is much more frequent in patients with pulmonary fibrosis secondary to ILD. Collected data are demonstrated in Table 2.

The change of the ultrasound frequency from 2 to 6 MHz leads to a shortening or even the disappearance of vertical artifacts (conversion to A lines was observed in three cases), and this phenomenon is more characteristic for pulmonary fibrosis than edema (61% vs. 24% of the examined areas, *p* < 10^−6^).

Differentiating the etiology of B-line artifacts (pulmonary edema versus pulmonary fibrosis) with the use of extreme frequencies (2 MHz and 6 MHz) is possible, both by detecting Z-lines at the frequency of 6 MHz (sensitivity 76%, specificity 62%, PPV 80%, NPV 56%, LR(+) 2.57, LR(−) 0.50), and by observing the change of the B-line profile to the Z-line when the frequency is increased (sensitivity 76%, specificity 61%, PPV 80%, NPV 55%, LR(+) 2.58, LR(−) 0.51, AUC= 0.687, *p* = 0.03). Statistically, both methods yield analogous results (to compare: AUC *p* = 0.67).

### 3.5. LUS Findings—Linear Transducer

Both groups (A and B), having been examined with a convex transducer, were re-evaluated with a linear transducer. Complete data obtained from 641 projections were analyzed statistically.

In patients with cardiogenic pulmonary edema, the pleural line was evaluated in 260 points. In 257 (98.8%) points, no pleural line abnormalities were detected. Only in three (1.2%) points were irregularities in the pleural line observed. Moreover, in 23% (60) of the evaluated areas, subpleural consolidations (up to 2–3 mm in diameter) were found in patients with cardiogenic pulmonary edema. These small consolidations correlated statistically significantly with vertical artifacts that, in the majority of cases, converted from B-lines to Z-lines when the frequency was changed from 2 MHz to 6 MHz in the convex probe (in 85%, *p* < 10^−6^).

In patients with pulmonary fibrosis, in all 381 points evaluating the pleural line, the following abnormalities were detected: coexisting irregularity and fragmentation of the pleural line in 68% (259 localizations) and blurred pleural line in 22% (84) of all evaluated points; the apparent thickening of the pleural line (>2 mm) was detected in only one (0.26%) area. Irregular and/or fragmented pleural line was present in 97% (230 out of 236) of the patients in the group in which B-lines converted into Z-lines (when the frequency was changed from 2 MHz to 6 MHz) (Figure 3).

These findings indicate that an irregular and fragmented pleural line is a feature that differentiates pulmonary fibrosis from cardiogenic pulmonary edema. Detection of this feature in a single evaluated point allows for diagnosis of pulmonary fibrosis with a specificity of 99%, a sensitivity of 68%, PPV 99%, NPV 68%, LR(+) 60, and LR(−) 0.32, at AUC = 0.836 and *p* = 0.0002.

An attempt at diagnosis: a differentiating model:

Although both features described above differentiate fibrosis from edema, a single observation point is not sufficient.

We proposed the best final model B, shown in Figure 4, as a decision tree graph. It is characterized not only by excellent Sp, Se, PPV, and NPV, but also the lowest Akaike criterion value. A low negative (<0.1) and high positive (>10) likelihood ratio indicates a high discriminatory value of the model. For example: suppose that a patient has an a priori probability of fibrosis of 50%. If the test result is positive for fibrosis, a posteriori probability increases to 96%, or else it falls to 9%. A comparative graph of the ROC curves tested is presented in Figure 5.

## 4. Discussion

### 4.1. Physical Hypothesis

It has been suggested in a previous paper [11,12] that every vertical artifact which can be observed in a LUS image is probably generated by multiple reflections between the walls of the lung aerated spaces. It is highly unlikely that a vertical artifact can be generated by a vibrating air bubble (alveolus or an alveolus partially filled with water) [12]. An acoustic trap is needed to generate a vertical artifact: (a) an acoustic pulse is transmitted from the thoracic wall to the trap through a thickened interstitial space; (b) multiple reflections between the walls of the aerated spaces which surround the trap generate an acoustic perturbation inside the trap; (c) such an acoustic perturbation acts as an ultrasound source, and gradually re-radiates the trapped acoustic energy to the transducer [12,13]. Figure 6 shows two types of acoustic trap. The panel on the left shows a medium (water, blood, tissue, etc.) which is connected to the thoracic wall by means of a single channel. The panel on the right shows a more complex acoustic trap, which is formed by sparse media connected to the thoracic wall by means of multiple channels. In the first case, the aperture of the acoustic trap is given by a single channel, while in the second case by multiple channels.

The characteristics which distinguish vertical artifacts are: brightness, length, lateral width, and internal structure. In this study, only the length of vertical artifacts has been analyzed, depending on the change in ultrasound frequency.

The length of an artifact is an interesting parameter, even though it represents really complex information. It depends on the duration of the trap response which, from a theoretical point of view, is infinite. Once a US pulse has been partially trapped by an acoustic trap, the latter re-radiates the trapped energy during an infinite time interval. Therefore, the question is: Why do we sometimes observe artifacts which reach the bottom of the screen and sometimes shorter artifacts? The answer lies in the signal to noise ratio (SNR). The amplitude of the trap signal decreases during a time interval until the signal is no longer distinguishable from the noise, and there are no possibilities for the time gain compensation (TGC) to make it visible. Therefore, now the question becomes: How much does the trap signal decrease during a time interval? This is an interesting question, and the answer should open our minds to the world of ultrasound artifacts. As an example, Figure 7 shows three different acoustic traps with a single channel aperture at the top. The left panel shows an almost closed trap. Once the acoustic energy is transmitted to this trap, the trapped energy escapes slowly, since there is an impenetrable air barrier everywhere except at the small aperture at the top. In this case, a long artifact is expected. The central panel displays a trap which is closed at the bottom and opened at the top. Once the acoustic energy is transmitted to the trap, the trapped energy meets an impenetrable barrier at the bottom and at the sides of the trap. However, a large aperture with a low impedance mismatch exists at the top which allows both: a minimal reflection (towards the bottom of the trap), and a significant transmission (towards the transducer) of the trapped energy. In this case, a short artifact is expected. The right panel shows a few traps with single apertures at the top, where traps are connected by secondary channels. Once acoustic energy is transmitted to one of these traps through a single aperture at the top, the trapped energy can escape the trap both through the aperture at the top (by providing in this way the artifact which is visualized by the transducer), and through the secondary channels. Due to the energy loss through the secondary channels, a shorter artifact is expected in this case also.

### 4.2. From the Physical to the Clinical Significance of the Study Results

The analysis of LUS findings in the case of diseases involving interstitial space presently poses a serious challenge for clinicians. In many diseases that affect the interstitium and alveoli, vertical reverberation artifacts are detected. B-line artifacts are found, for instance, in: cardiogenic pulmonary edema, noncardiogenic pulmonary edema, and ILD (both in the active phase, when ground-glass opacities are found in HRCT, and in the fibrotic phase, in which honeycombing is the corresponding HRCT finding) [10,14,15,16,17]. Z-line artifacts have not yet been described as having clinical significance.

So far, the literature in the diagnosis of pulmonary fibrosis has been based mainly on B-line artifacts [18]. It has been proved, inter alia, that the more severe the pulmonary fibrosis, the more B-line artifacts are visible on lung ultrasound images [19,20,21,22]. The publications also emphasized the importance of coexistence with B-line artifacts of lesions in the pleural line [23,24,25,26,27,28]. Consideration should be given here to the diversity of the study protocols, depending on the investigator. The use of convex and linear probes allows for a general and detailed assessment of lesions on the pleural line and the subpleural area [28]. This shows how technical aspects can influence the quality of the ultrasound examination. Optimizing the settings of the ultrasound machine becomes another link in improving the quality of ultrasound in the diagnosis of diseases that affect the interstitial space of the lungs.

### 4.3. Pulmonary Fibrosis

We observed a change in the length of vertical artifacts in both examined groups, and the concurrent presence of pleural line abnormalities. In the first examined group (A), conversion of B-lines to Z-lines was usually accompanied with pleural line abnormalities and/or small subpleural lesions. In this patient group, both B-line to Z-line, at two extreme frequencies (2 MHz and 6 MHz) this most likely results from generating artifacts in acoustic traps, that are adjacent to the pleural line, by means of a single large channel and multiple channels. Probably both types of traps are present at the pleural line, given the apparent shape of the pleural line, which appears irregular and blurred.

### 4.4. Pulmonary Edema

Changes that occurred during the conversion of B-line artifacts in the second patient group (B) were much more diversified. A large number of B-lines did not undergo conversion with a change in ultrasound frequency. Concurrently, some B-line artifacts converted to Z-lines, or B- and Z-lines. However, a correlation between the conversion of B-line artifacts to Z-lines (or B- and Z-lines) and coexisting pleural line abnormalities was observed in this group as well. This was most likely associated with the presence of microconsolidations (for instance, due to small areas of atelectasis caused by pulmonary edema).

This study also demonstrates that differentiating a cardiogenic etiology of B-line artifacts from a pulmonary etiology should be based, both on the analysis of the artifact length after the conversion, and on the evaluation of pleural line abnormalities. Accounting for both findings results in a much higher accuracy in differentiating the causes of lesions affecting the insterstitium.

### 4.5. Study Limitations

The only analyzed feature was the artifact length. It is possible that more factors (brightness, lateral width, internal structure) are significant in differentiating the etiology of vertical artifacts; however, these factors were not analyzed in this study.

The second limitation is the selection of patients for the study group. Patients with chronic pulmonary congestion, but in a stable clinical condition, were not included in the study.

The third limitation is given by the lack of information on the pulse power spectrum and on the modulation transfer functions of the probes.

## 5. Conclusions

The visualization and sonomorphological analysis of vertical artifacts with the application of a convex transducer employing two different, extreme frequencies (2 MHz and 6 MHz) may be useful in differentiating lesions affecting the interstitium (cardiogenic pulmonary edema vs lesions due to ILD). However, a higher accuracy is achieved when the pleural line and subpleural lesions are concurrently evaluated with a linear transducer.

## Figures and Tables

**Figure 1 diagnostics-11-00401-f001:**
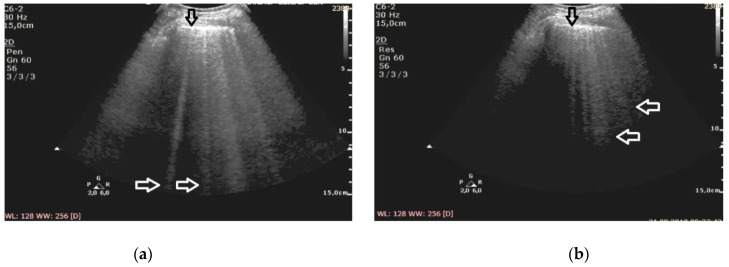
Pulmonary fibrosis; (**a**) B-lines at 2 MHz (white arrows), irregular pleural line (black arrow), (**b**) Z-lines at 6 MHz (white arrows), irregular pleural line (black arrow).

**Figure 2 diagnostics-11-00401-f002:**
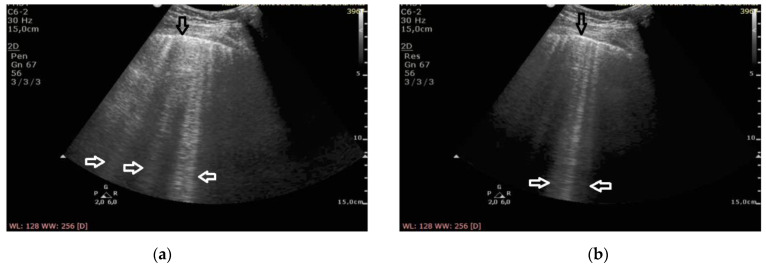
Cardiogenic pulmonary edema; (**a**) B-lines at 2 MHz (white arrows), regular pleural line (black arrow), (**b**) B-lines at 6 MHz (white arrow), regular pleural line (black arrow).

**Figure 3 diagnostics-11-00401-f003:**
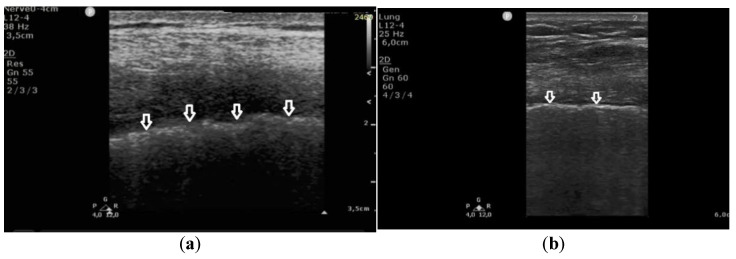
Linear transducer used for the evaluation of pleural line abnormalities: (**a**) irregular, blurred pleural line (white arrows), coarse in appearance, in pulmonary fibrosis; (**b**) regular pleural line, with preserved echogenicity (white arrows), in pulmonary edema.

**Figure 4 diagnostics-11-00401-f004:**
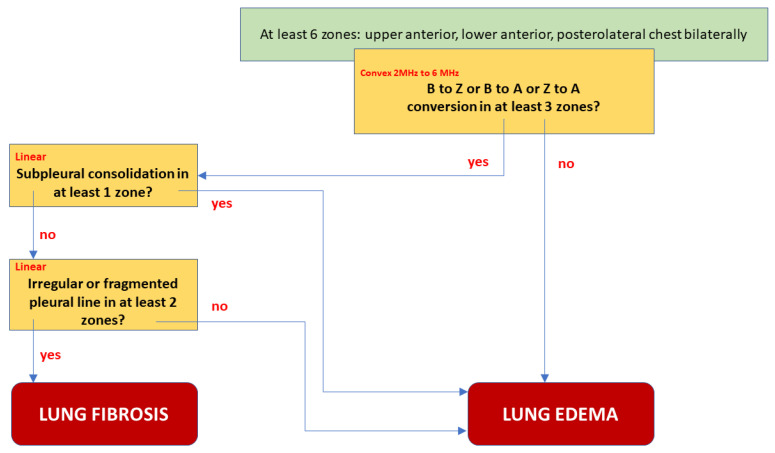
Differentiating between pulmonary fibrosis and cardiogenic lung edema. Decision tree corresponding with model B (in the text).

**Figure 5 diagnostics-11-00401-f005:**
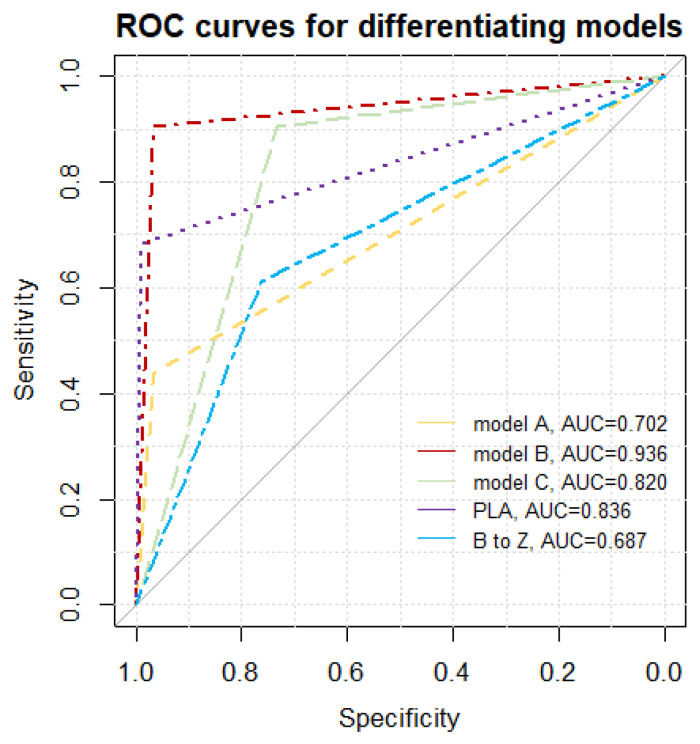
Receiver operating characteristic (ROC) curves for methods of differentiation between pulmonary fibrosis and pulmonary edema. The area under the curve for model B is significantly greater than for other models, with maximal *p*-value of 0.0033 (model B compared to PLA); PLA—pleural line abnormalities in a single examination zone, B to Z—Conversion of B-line artifacts to Z-line (with the frequency changed from 2 MHz to 6 MHz) in a single examination zone; model A—any conversion of vertical artifacts (B to Z, B to A, Z to A with the frequency changed from 2 MHz to 6 MHz) in 3 or more areas, and lack of consolidations, model B—any conversion of vertical artifacts (B to Z, B to A, Z to A with the frequency changed from 2 MHz to 6 MHz) in 3 or more areas, lack of consolidations, and PLA in at least 2 zones, model C—any conversion of vertical artifacts (B to Z, B to A, Z to A with the frequency changed from 2 MHz to 6 MHz) in 3 or more areas, and PLA in at least 2 zones. AUC—area under the curve.

**Figure 6 diagnostics-11-00401-f006:**
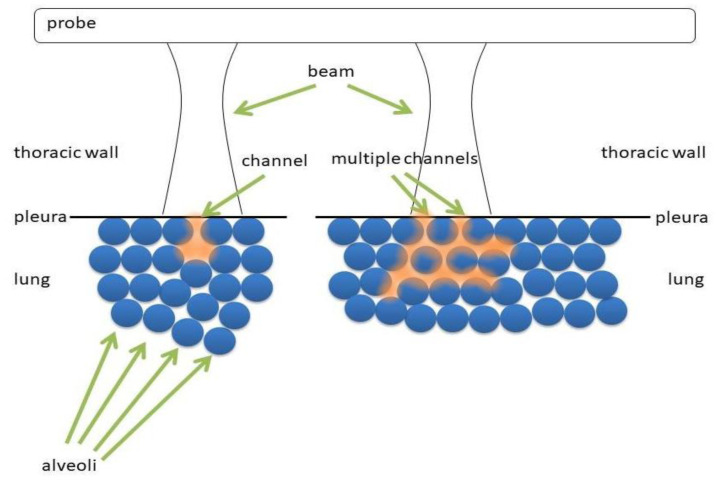
Two types of access channels to the underlying acoustic traps for ultrasound beams.

**Figure 7 diagnostics-11-00401-f007:**
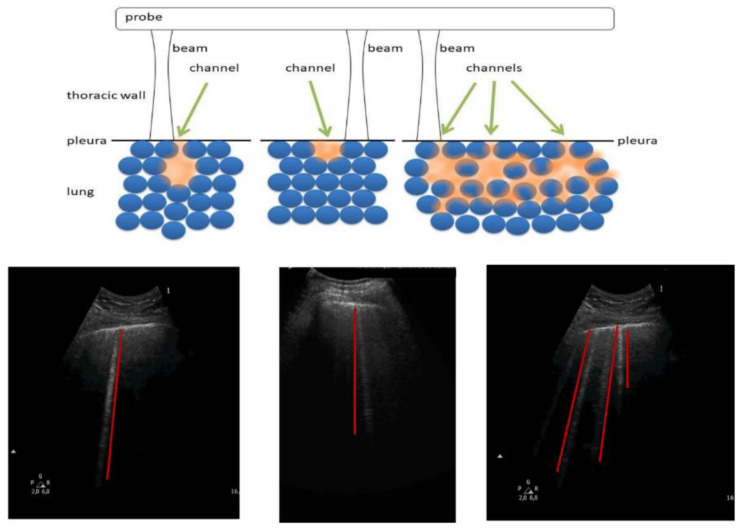
Relations between the length of a vertical artifact and the shape of the acoustic trap.

**Table 1 diagnostics-11-00401-t001:** Sensitivity and specificity of findings differentiating pulmonary edema from pulmonary fibrosis in interstitial lung disease (ILD). PLA—pleural line abnormalities, B to Z—Conversion of B-line artifacts to Z-line (with the frequency changed from 2 MHz to 6 MHz), *—single examination zone; SE—sensitivity, SP—specificity, PPV—positive predictive value, NPV—negative predictive value, LR(+)—positive likelihood ratio, LR(−)—negative likelihood ratio, AUC—area under the curve, AIC—Akaike information criterion, OR—odds ratio, CI—confidence interval.

Finding	SE (%)	SP (%)	PPV (%)	NPV (%)	LR(+)	LR(−)	AUC	*p*-Value	AIC	OR (95% CI)
PLA *	68	99	99	68	60	0.32	0.836	0.0002	510.8	184.08 (57.79–586.28)
B to Z *	61	76	80	55	2.58	0.51	0.687	0.03	779.8	5.03 (3.54–7.15)
model A	44	97	93	62	13.13	0.58	0.702	0.035	73.91	22.56 (2.73–186.47)
model B	91	97	97	91	27.19	0.097	0.936	7 × 10^−^^6^	32.68	280.33 (27.52–2855.45)
model C	91	73	78	88	3.40	0.13	0.820	0.0012	60.98	26.58 (6.31–111.97)

**Table 2 diagnostics-11-00401-t002:** Conversion of vertical artifacts depending on the set ultrasound frequency.

Conversion of Vertical Artifacts with the Change of the Frequency from 2 MHz to 6 MHz	Pulmonary Edema (*n* = 250)	Pulmonary Fibrosis (*n* = 394)	*p*
B to B and Z	41 (16%)	41(10%)	<10^−6^
B to Z	18 (7%)	193 (49%)	
B to non-verticals (A)	0	1 (<1%)	
B and Z to Z	0	3 (<1%)	
A to B	1 (<1%)	0	
Z to non-verticals (A)	0	2 (<1%)	
No change	190 (76%)	154 (39%)	

## Data Availability

Data sharing not applicable.

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
