# Peer review of "Clinical Impact of Vertical Artifacts Changing with Frequency in Lung Ultrasound"

_diagnostics, 2021, doi:10.3390/diagnostics11030401_

Round 1

Reviewer 1 Report

The work is very interesting and provides most of the explanations necessary to fully understand the physical phenomena underlying the diffusion of ultrasounds in a pathological lung. The possibility of distinguishing the B lines in the various pathologies of the interstitium is a necessity to raise the thoracic ultrasound technique to the level it deserves and this work continues in the vein of this research. Having proved that the presence and characteristics of vertical artifacts depend on the frequency of the ultrasound beam and its interaction with certain physical and spatial characteristics of the medium it hits is an important step forward that must be recognized to the authors.

Please only correct a sentence without a conclusion in paragraph 2.5 at point B and provide the required keywords, the presence/absence of fundings for the research, conflict of interset and informed consent statement, data availability and eventual aknowledgemnts.

Author Response

Dear reviewer
Thank you for your positive review and comments.
Corrections have been made in the following manuscript (marked in yellow in the text).
Keywords are included, the presence of funds for the research, conflict of interset and informed consent statement, data availability and eventual aknowledgemnts

Reviewer 2 Report

Very interesting and valuable work. The article is well prepared. The presented algorithm of differentiation pulmonary edema from pulmonary fibrosis is clinically useful. 

Author Response

Dear reviewer
Thank you for your positive review and comments.
In the following manuscript I add key words, sourse of founding and conflict of interest (marked in yellow in the text).
